# Microstructural Analysis of Organic-Rich Shales: Insights from an Electron Microscopic Study by Application of FIBSEM and TEM

**DOI:** 10.3390/nano12234135

**Published:** 2022-11-23

**Authors:** Jinxuan Han, Hongjian Zhu, Yanjun Lu, Su Yang, Manping Yang, Erxiu Shi, Yu Qi

**Affiliations:** 1School of Vehicle and Energy, Yanshan University, Qinhuangdao 066000, China; 2State Key Laboratory of Continental Dynamics, Northwest University, Xi’an 710069, China; 3Petroleum Engineering Technology Research Institute, Sinopec Southwest Oil and Gas Company, Deyang 618000, China

**Keywords:** FIBSEM, TEM, marine shale, microstructure, pore, microfracture

## Abstract

Matrix-related pores play a significant role in controlling hydrocarbon production in organic-rich shales. Multiple matrix-related pore types of typical marine shales in the Sichuan Basin have been visually investigated and identified by transmission electron microscopy (TEM) on ultra-thin sections and by focused ion beam-scanning electron microscopy (FIBSEM) on polished sections. OM-hosted pores seem universal and range in sizes from below 1 nm to hundreds of nanometers and they are not homogeneously developed and distributed, which is mainly determined by thermal maturity and OM composition. Mineral-hosted pores are defined by mineral frameworks and occur in open spaces related to ductile or rigid grain fabric. The four porous mineral types that occur are clay intrapores, carbonate solvopores, pyrite interpores, and quartz interpores, and they range in size from less than 1 nm to more than several microns. Aggregate-hosted pores are predominantly associated with clay-organic aggregates, pyrite-organic aggregates, clay-pyrite aggregates, and clay-organic-pyrite aggregates. The most common aggregate-hosted pore networks are defined by clay-organic aggregates, and the pores are largely developed between the clay and organic layers and may be the important adsorption spaces for methane. Fracture-related pores include microchannels and microfractures of various sizes and shapes and they could play a key role in providing hydrocarbon migration pathways. FIBSEM and TEM show direct evidence that OM-hosted pores and fracture-related pores contribute more to the effective pore network and the excellent reservoir quality, whereas poor reservoir quality may come from aggregate-hosted pores.

## 1. Introduction 

In recent years, the Lower Cambrian and Lower Silurian shales have been the most actively drilled shale gas deposits in the South China [1,2,3,4]. Shale gas production from these organic-rich formations makes up an ever-increasing percentage of total gas production in China, which is a significant response to the ever-increasing demand for alternatives to conventional oil and gas energy [5,6,7]. For example, recent shale gas explorations in the Sichuan Basin, Xichang Basin, and Yanyuan Basin in Sichuan province from a total of 11 organic-rich shale formations in 2022 have technically recoverable reserves at 9 Tcf (trillion cubic feet), with economically recoverable-reserves of 5 Tcf. A great deal of research has been significantly reported in the literature about petrology, mineralogy, organic geochemistry, reservoir quality, generation potential, petrophysical properties, and microstructures in low-porosity and low-permeability shales [2,3,4,8,9,10,11,12,13,14]. The pore system preserved in fine grain, organic-rich shale can be used to identify gas storage space and migration pathways and estimate the total gas resources, thus serving as a distinct signature of predicting how hydrocarbon molecule are adsorbed and how transport takes place in such porous medium because the morphology, size, continuity, and occurrence medium of the pores can provide insights [4,15,16,17,18,19,20]. 

Some attempt analyses such as helium porosity (HP), low-pressure gas adsorption (LPGA), mercury injection pressure (MIP), high-pressure CH_4_ isotherm analysis, and nuclear magnetic resonance (NMR) spectroscopy have been used to assess pore structures (morphology, size, connectivity, etc.) [5,6,7]. These techniques do not involve visual description and direct observation of shale pore systems. Therefore, they were not used in this study. The electron microscopy (EM) technique has long been in common usage in natural ultra-fine-grained rocks such as mudstones, shales, coals, and tight sandstones and their constituent OM and clay minerals. As suggested by Loucks et al. [15,16], the vast majority of pore spaces in shales or mudstones are commonly smaller than a few micrometers in size and cannot be observed using light microscopy and low-resolution SEM. These workers also pointed that if the surface of the samples is not prepared using ion-milling techniques, the small pores could not be identified using SEM/TEM-based imaging techniques. To obtain better images of the micro- to nanometer-scaled structures of the shale samples and thus understand their effects on gas storage and transport, high-quality preparation (e.g., argon ion beam milling and FIB slice extraction techniques) and imaging techniques (e.g., SEM and TEM) incorporated onto the same sample chamber are significant [3,14]. These analyses and images allow for the identification of the type, shape, size, location, continuity, and abundance of the pores in 2D and 3D. SEM of FIB cross-sectioned sample surfaces is a major powerful technique, as secondary electron and backscattered electron SEM imaging (SE-BSE) combined with qualitative energy-dispersive spectroscopic (EDS) analysis is used to characterize the evolution of the mineralogical and organic structures and their related pore space in shales or mudstones, which is very successful in imaging micro- to nanometer-scaled fabric signatures down to ~100 nm [15,16,17,18]. However, many published data based on indirect petrophysical methods (i.e., HP, LPGA, MIP, and NMR) suggest there is a large pore size contribution from <100 nm features that are not accounted for in SEM methods [8,9,10,11]. By contrast, direct imaging by TEM is capable of resolving such fine structures because it can provide resolutions below 0.2 nm. The TEM tool, which uses intact ion-milled specimens, allows organic/inorganic constituents and their related pore structures to be imaged in situ, providing detailed characterization of morphology, composition, and crystallographic features. More importantly, TEM can provide definitions of the nanometer-scaled structural data, most of which have sizes beyond that of resolution by SEM. Therefore, SEM and TEM collectively contribute to the characterization of the complex micro- to nanometer-scaled structures and give new insights into how such pore systems store and transport gas within organic-rich shales [3,14]. 

Previous studies have offered further evidence for a variety of microscale pores developed in shale [15,16,17,18]. However, detailed, visualized investigations of the shale nanopores are still rare. The present study steps into this gap and compiles data gained from applied integrated FIBSEM and TEM 2D imaging and provides visual qualitative analyses of the organic-rich shale reservoir pore system at a micro- to nanoscale. The primary goals of this study include: (1) demonstrating the FIBSEM and TEM microanalysis capabilities on organic-rich shale samples, (2) providing qualitative data on mineralogical and organic components, fabrics, and their related aggregates, (3) determining the in situ distribution and signatures of the organic/inorganic pore networks at a micro- to nanoscale, (4) presenting a pore classification that summarizes the spectrum of pore types based on SEM/TEM data, (5) discussing how various pore types contribute to effective gas storage space and matrix-flow pathways, and (6) gaining insights into the control of organic porosity across a range of thermal maturity. 

## 2. Materials and Methods

### 2.1. Samples 

The shale samples investigated in this study come from cores (wells of JY1 and Shallow HY1) and well-exposed outcrops collected from the Lower Cambrian Qiongzhusi (or Lujiaping) and Lower Silurian Longmaxi shales in the Sichuan Basin, South China. The Sichuan basin and its peripheral regions are the important natural gas and shale gas players in China and are located mainly in the Upper Yangtze Block. The Lower Cambrian and Lower Silurian marine strata are widely distributed over this area and have been recognized as effective source rocks, thus indicating huge shale gas resources potential. The early Cambrian Formation generally developed siliceous deep-water shales due to rapid sea-level rises, and the lower Cambrian marine strata are the high-quality reservoirs being studied for shale gas exploration in the Upper Yangtze region, South China. The thickness of the Silurian Longmaxi shale ranges from 50 to ~500 m with the total organic carbon (TOC) content of 1.0~4.0 wt.% and the *R*_o_ of 2.0~3.0% [1,11,19]. Such a formation was deposited in a low energy anoxic sedimentary environment and can be divided into three lithologic sections from bottom to top: (1) black organic-rich siliceous shale, (2) gray-dark carbonaceous shale, and (3) dark-gray mudstone. Both shale formations with abundant organic matter and large thickness are currently being explored as shale gas reservoirs, hence an investigation of microstructural features is significant for understanding gas storage and transport, as well as for estimating gas capacities and booking reserves. 

### 2.2. FIBSEM Method 

The FEI Helios Nanolab^TM^ 600 DualBeam™ microscope operating at the research center of analysis and measurement of Harbin Institute of Technology (Harbin, China) has been used to perform 2D FIBSEM investigations. Dual Beam FIBs are a type of instrumentation that include a SEM column with a fine-probe ion source. The Helios NanoLab™ 600 is equipped with a high-resolution Elstar™ electron column with a field emission gun (FEG) electron source. This instrumentation is capable of imaging with a resolution better than 1 nm at 15 kV and 2.5 nm at 1 kV beam energies. The Gallium ion beam (Tomahawk™ ion column) can image and remove material down to 5 nm resolution levels. This instrument is capable of preparing SEM samples from specific areas of a shale sample as well as nano-machining and 2D/3D imaging. In this study, we used FIBSEM to observe 2D microstructural characteristics (more than 100 nm). The shale samples were mounted to SEM stubs on a small block of about 1 cm^2^ in size. These broken samples were polished to produce a level surface using dry emery paper and were then milled by an argon-ion method. After polishing, the samples were coated with gold to provide a conductive surface layer. Therefore, the 2D microstructures of the polished sample surface can be obtained by opening the electron beam and closing the ion beam. A detailed description of the theory and method of the FIBSEM technique can be gained through reference to Curtis et al. [21], Bernard et al. [22], and Zhou et al. [23]. 

### 2.3. TEM Method 

Transmission electron microscopy (TEM) has been widely applied in the fields of energy, biological and material science, and engineering. It has been increasingly applied in the fields of geological rocks to characterize the micro- to nanometer-scaled features of quartz, clay, and organic matter, as well as their related pore structures. Significantly, TEM significantly offers a powerful alternative to visualize some nanostructures whose characteristic dimensions are less than 100 nm in size. This technique can provide resolutions below 0.2 nm and offer qualitative and quantitative characterization of nanostructures and their composition. In addition, as a very powerful tool, TEM allows for the observation of relatively thin samples because the quality of TEM images significantly depends on the quality of the samples. In general, the samples must be sufficiently thin, about 100~500 nm. A selected area was prepared using an FIB slice extraction technique. The ultra-thin sections were mounted on a TEM grid placed in a TEM for analysis. The TEM technique has been used as a complementary method to other prominent techniques such as SEM, thin section and light microscopy, CT, AFM, etc. It can improve our understanding of the nanostructural characteristics of organic-rich rocks, such as coal and shale. TEM was performed on ultra-thin sections at Peking University (Beijing, China) using a Tecnai F20 XTWIN TEM operated at 200 kV with a field emission gun as the electron source. For an extensive, detailed description of the theoretical and practical background of the TEM technique, one can refer to Goodhew et al. [24]. In addition, the TEM sample preparation techniques and the application of TEM for coals and shales have been published by Bernard et al. [22] and Kwiecińska et al. [25]. 

Note that FIBSEM and TEM can well visualize the shale microstructures, but they have some limitations in pore quantification, field representation, and solving the heterogeneity of shale microstructures. In recent years, many scholars have applied a variety of auxiliary software, such as Image J, to apply its image processing function to FIBSEM and TEM images, so as to obtain some semi-quantitative data based on visual data [14]. Compared with LPGA and MIP experiments, these data can better capture the microstructural information of some closed pores. It is undeniable that the EM technique has led to the fine characterization technique of the micro- to nanostructures of tight rocks such as shale and mudstone.

## 3. Results and Discussion

### 3.1. FIBSEM/TEM-Based Shale Pore-Type Classes

Pore networks within shales range from simple to complex in morphology, abundance, and origin, with most pores being less than several microns. Several publications have provided quantitative data and excellent images of shale or mudstones pores acquired by LPGA, MIP, NMR, and SEM petrographic evidence [8,9,14,15,16,17,18]. On the basis of the classifications of the International Union of Pure and Applied Chemistry (IUPAC), the pores that were obtained fell into three classes: micropores (pore width < 2 nm), mesopores (pore width between 2 and 50 nm), and macropores (pore width > 50 nm). Slatt and O’Brien [18] published a classification scheme for Mississippian Barnett and Devonian Woodford shales that more considerably considered the gas storage space and migration pathways. Moreover, Loucks et al. [16] were the first to suggest a systematic classification of pore types associated with mudrock systems, recognizing mineral matrix pores, OM pores, and fracture pores. The classification clearly focuses on OM pores as the major connected or effective pore network in some shale gas systems, such as the Barnett shale. In addition, this classification is simple and objective and incorporated some inherent characteristics of rocks, such as porosity, permeability, wettability, and connectivity. There is currently no other better classification scheme for mudrock pore systems than this. Therefore, this classification has been widely used in most recent research. In these classifications, these pore types are formed by both depositional and diagenetic processes and may be significantly influenced by compaction, cementation, dissolution, and organic–inorganic reactions, as well as later tectonism. On the basis of these classifications, other authors presented some updated classification schemes, which dealt more with the inorganic–organic association and structural deformation. Zhu et al. [14] suggested and demonstrated that it is important to value these two processes in the shale pore type classification. 

Abundant micro- to nanometer-scaled structures of organic/inorganic materials and their related pore networks are developed in shale, which are varied and comprise multiple pore types with variable sizes, morphologies, connectivities, and composabilities. Note that conventional light microscopy, thin sections, HP, LPGA, MIP, and NMR often fail to adequately recognize and qualitatively characterize these pore networks, especially nanopores. However, a combination of FIBSEM and TEM can make up for the shortcomings of these technologies. FIBSEM and TEM observations suggest that these nanostructures are all large enough to store a large quantity of hydrocarbon molecules and most likely contribute to permeability at a nanoscale. Consequently, the FIBSEM and TEM analyses were performed on the shale samples. Four universal pore types exist: (1) OM-hosted pores, (2) mineral-hosted pores, (3) aggregate-hosted pores, and (4) fracture-related pores. 

#### 3.1.1. OM-Hosted Pores 

OM-hosted pores are open intrapores that are developed within OM with isolated, irregular, bubble-like, elliptical cross sections. They generally range from several to hundreds of nanometers in length and have been well documented since they were first observed in the Barnett shale by SEM [15]. Positive relationships, such as between the pore structure parameters and total organic carbon (TOC) contents, as well as between organopores and OM, suggest that OM-hosted pores display good connectivity and can be important contributors to the shales or mudstones’ pore networks. In addition, some early studies on organic porosity observed in mudrocks indicate that OM-hosted pores have proven to be significant for hydrocarbons production from organic-rich reservoirs [15,16,17,18]. For example, Ji et al. [26] investigated the OM types and associated nanopores in highly mature gas Longmaxi shales in the upper Yangtze platform, South China. They indicated that the pore systems are dominated by OM-hosted pores that can form effective pore networks. 

Based on SEM observation, Ko et al. [27] identified two significant organic pore types in the Eagle Ford Group shale samples, south Texas Upper Cretaceous shelf: primary and secondary OM pores, which include OM bubble pores and OM spongy pores. They suggested that the abundance of OM spongy pores correlates positively with TOC content. The Lower Cambrian Qingzhusi and Lower Silurian Longmaxi formations are the most prolific shale-gas resource players in the Sichuan Basin, south China. Previous studies have suggested that OM pores are the most common and dominant pores in these shales [1,10,11]. OM domains vary widely in size and form. At lower magnification in SEM images, OM particles can be up to a few microns in width and hundreds of microns in length. In addition, at higher magnification in both SEM and TEM images, OM particles can be down to a few nanometers in width and hundreds of nanometers in length. 

Three OM occurrence types exist: (1) depositional in-place OM, (2) migrated OM, and (3) OM-mineral aggregates. The type of OM-mineral aggregates and associated pore networks will be addressed in the following sections. Based on the morphologic features, Schieber [28] presented a descriptive classification of OM in the New Albany core, which categorizes OM into two major types, including structured organic matter (SOM) and amorphous organic matter (AOM). Furthermore, Loucks and Reed [29] successfully identified the nanometer- to micrometer-sized OM types using a SEM. They put forward seven petrographic criteria for separating depositional versus migrated OM on the basis of the organic pore characteristics and their relationships to grains. This OM type classification has been widely used in recent works and fits the definition of the OM occurrence types used in this paper. They also suggested that distinguishing OM occurrence types is very important because the proportions of these different OM domains could cause the final OM-hosted pore systems and associated porosity and permeability. The dominant type of OM domains seen in the samples is secondary migrated OM with flow characteristics, which present ribbon, arcuate, and dispersed outlines (Figure 1). Such OM particles can migrate into microfractures (Figure 1A) and fill the large interpores (Figure 1B–E) and grain-rim dissolution pores (Figure 1F) as liquid hydrocarbons or bitumen with abundant pore development. They commonly elongate parallel to the bedding and squeeze around or cut rigid grains. The OM particles that exist between the rigid grains show a low degree of compaction. Another very important type of OM domain is nonmigrated OM (i.e., depositional in-place OM) (Figure 2), which is isolated and lacks distinct outlines and sizes with an inhomogeneous texture. The OM-hosted pores comprise both isolated bubble pores and irregular spongy pores on the basis of their morphology and connectivity in both depositional in-place OM and migrated OM (Figure 1 and Figure 2). These are not the new types of organic pores. Instead, both are the most important but very common pore type in organic-rich shales. OM isolated bubble pores are related to the thermal maturity, whereas OM irregular spongy pores are the result of a combination of the OM original structure and maturation. Additionally, mineral sheltering could significantly protect these organic pores from compaction in a local micro-zone. 

OM isolated bubble pores generated during burial and thermal maturation of OM can range from dozens of nanometers to hundreds of nanometers in size and they are most commonly observed in both non-migrated and migrated OM. These bubble pores usually display isolated and rounded to subrounded outlines in 2D (Figure 1B,F and Figure 2A,B,D–F), which seems to have a poor connectivity if we not consider their spatial distribution in 3D. Therefore, their contribution to matrix-flow permeability remains speculative. The TEM images allow the observation of the inner surface of nanopore walls, with most of the indentations (i.e., smaller pores) measuring approximately 1 nm in size. These abundant smaller pores may create such irregular surfaces in larger pore walls [17]. Thus, most pores present complex internal nanostructures and spherical inner walls with highly irregular surfaces, which significantly suggests that small pores could provide higher surface areas and greater adsorption energies compared to large pores [3]. Some large OM isolated bubble pores are marked with black dotted lines in Figure 2E,F, and the numbers around the pores are the pore diameters that we estimated. It can be seen from Figure 2E,F that the pore diameter of these large pores is less than 100 nm, concentrated in 60 to 90 nm. It is worth noting that some smaller pores cannot be identified due to the TEM image resolution, but it can be inferred that the actual pore sizes of OM isolated bubble pores are much smaller than what we observed. 

OM irregular spongy pores are connectable and have a subrounded or subangular outline in 2D (Figure 1 D,E and Figure 2C), which seems to have a good connectivity in 3D. OM irregular spongy pores vary greatly in pore size. TEM observation results show that a porous organic matter particle can hold thousands of pores, which have different shapes, and the largest pore space is only 10 nm (Figure 1D,E). In contrast, the FIBSEM image shows that these pores sizes range from tens of nanometers to hundreds of nanometers (Figure 2C). These pores also show complex internal nanostructures and rough inner walls. Such shapes of the pores suggests that they undergo further growth after they are formed. Several adjacent pores can merge, thus producing convolute pore shapes. Loucks et al. (2012) suggested that the OM irregular spongy pores are most commonly found in depositional in-place OM and may be produced by the gas generation at higher stages of maturation. 

#### 3.1.2. Mineral-Hosted Pores

Mineral-hosted pores are defined by mineral frameworks and occur in open spaces related to ductile or rigid grain fabric. Four porous mineral types can be identified including clay intrapores, carbonate solvopores, pyrite interpores, and quartz interpores. 

(I)Clay Intrapores

Visual estimates from many FIBSEM and TEM images have indicated that mineral-hosted pores defined by clay domains are the most ubiquitous pore type in shale samples. The shape and size of clay pores depend strongly on the occurrence and accumulation of clay particles (Figure 3). If the clay domains are loosely packed with little compaction or produced within the large interpores, a significant structure of “cardhouse” can be produced due to individual edge-to-face-, edge-to-edge-, or face-to-face-oriented clay flakes (Figure 3A,B) [18]. Such structures provide abundant intrapores within the clay mineral platelets that have lengths of about 50 to 500 nm (Figure 3A). These pores are commonly scattered and indicate little to no preferential orientation except in localized areas. In some localized micro-zones of the samples, clay intrapores are typically reduced in abundance and size by compaction (Figure 3C–E, indicated by orange arrows). Slit porosity are aligned parallel with laminae except a few subvertical pores. Most clay linear pores observed by FIBSEM have lengths in the 5 μm range and have widths in the 1 μm range but they can range in size from 50 nm to several micrometers in length and from several nanometers to 500 nm in width (Figure 3C,D). The size of such clay intrapores observed by TEM is much smaller (Figure 3E). The length of these pores does not exceed 50 nm; the width is only within 2 nm, which is in the range of micropores. Zhu et al. [14] observed the similar clay ultrastructure of clay-rich shale within an illite domain with TEM. These stacked tiny pores with pore sizes measuring below 1 nm belong to the linear openings and are primary in origin. They suggested that these tiny, linear clay intrapores may not produce important gas permeability pathways and storage space within the clay matrix. Note that a large number of clay intrapores occur where more ductile clay flakes bend around a rigid grain (e.g., quartz) (Figure 3F,G, indicated by orange arrows). These pores can be preserved because the rigid grains could form a sheltering effect, which may strongly inhibit the compaction between the ductile clay grains. Whereas the sheltering effect is most helpful for preserving clay intrapores, clay sheets clamping can also provide protection from compactional closure [28]. 

FIBSEM and TEM images of Figure 3 show some excellent examples of this phenomenon. Such pores found within clay aggregates consist of sigmoid-shaped or elongated openings that are defined by the randomly oriented clay flakes, ranging in size from less than 1 nanometer to more than hundreds of nanometers in size. In addition, some slit-shaped pores can be observed as interparticle porosity along the grain boundaries between clay sheets and some rigid grains (Figure 3A,F,G). The significance of these observations is that the flakes can provide abundant open pore networks within them. Most clay intrapores found by FIBSEM and TEM are larger than the 0.38 nm size of CH_4_, except for some micropores in a clay domain, and they can be interconnected to form significant CH_4_ storage space and permeability pathways. Moreover, within clay particles’ aggregates, clay convex nanostructures (i.e., lath-like illitic) are exposed within the large clay intrapores (Figure 3C,D,F,H,I, indicated by green arrows), and such nanostructures can provide a large number of methane adsorption sites. 

(II)Carbonate Solvopores

Dissolution pores (i.e., solvopores) are commonly observed within or around typically carbonate grains, such as dolomite and calcite, and thus produce significant intrapores (Figure 4, indicated by orange arrows) and grain-rim interpores (Figure 4, indicated by green arrows). In our samples, the dissolution rarely removes the entire carbonate grains. On the contrary, partial dissolution is more commonly observed within or along the carbonate grain margins, producing an open pore network in carbonate minerals. The FIBSEM and TEM analyses of the samples illustrate a large volume of channel and vuggy macropores (pore width > 50 nm) (Figure 4A,C–E) and mesopores (pore width 2–50 nm) (Figure 4B), respectively, which are the products of carbonate partial dissolution. The solvopore size observed by FIBSEM is large, generally concentrated in 300 to 500 nm (Figure 4A,C–E). Pores with pore diameters less than 100 nm cannot be clearly observed, and some macropores with a pore diameter of more than several microns are also widely developed (Figure 4C). In contrast, the pore size observed by TEM is very tiny, and most of the solvopores are within 50 nanometers (Figure 4B). It is worth noting that the pore size of intraparticle solvopores is often larger than that of grain-rim pores, but the connectivity of grain-rim pores may be greater, which is more worthy of attention by oil and gas geologists. A subhorizontal large solvopore through a carbonate grain exposes abundant nanopores developed at the inner surface of pore walls (Figure 4E). These nanopores have irregular, elliptical cross sections as a result of the structure of the carbonate substrate. Such pores are related pores produced by the partial dissolution of a grain. However, the crystal-mold pores first identified by Loucks et al. [16] in the Lower Cretaceous Pearsall Formation are not seen in these samples. It is commonly suggested that crystal-mold pores are produced by the complete dissolution of the carbonate minerals. 

As a common type of secondary porosity in the shales, carbonate solvopores can provide significant gas storage space and permeability pathways. In addition, the high porosities observed in the carbonate minerals imply that bulk carbonate densities are low and thus strongly influence not only the estimation of bulk volume of carbonate from weight percent of carbonate but also the later hydraulic fracturing. This is because carbonate mineral is a very important brittle mineral, in addition to quartz. 

(III)Pyrite Interpores

Pyrite framboids and crystals are locally common in the samples (Figure 5). Pyrite framboids consist of many small pyrite crystals, between which abundant interpores occur (Figure 5A,B). These pores are sharply angular in shape, with the pore size mostly larger than 50 nm. Note that pyrite interpores vary with the size of the framboids and crystals. Smaller framboids form smaller pores, whereas larger framboids show larger pores, and this finding has been included in many studies. Interpores can also be created between large pyrite crystals and other rigid grains (50~300 nm in size, Figure 5C,D). These Pyrite framboids and associated interparticle nanopores are consistent with observations made by Louck et al. [15] on mudstones from the Barnett Formation using the SEM. They found abundant pyrite-framboid intercrystalline pores in these mudstones, which importantly contribute to gas storage. Pyrite-hosted pores are well-developed in the shale matrix, but its pore network and pore volume are relatively small, which is mainly due to the three following reasons: (1) the pyrite content in shale is limited, generally about 5 wt.%, and such a small proportion of composition is not enough to make such pores become the main microstructure, (2) pyrite is an unstable mineral, which is easily affected by weathering, and in particular, the content of pyrite in shallow and outcrop shale is very low, and most particles are lost due to weathering, and (3) interpores within pyrite framboids are often filled or occluded with transported OM. OM fillings commonly break up larger interpores into smaller pores. 

(IV)Quartz Interpores

Some quartz interpores show limited porosity by using SEM imaging (Figure 6A). We speculate that quartz interpores could be easily affected by compaction and cementation, which makes them impossible to observe at lower magnification. Quartz interpores can be partly or completely occluded with OM or with authigenic clay platelets (Figure 6A). Combination of compaction and cementation significantly breaks up larger quartz interpores into smaller pores or closed pores. In some micro-areas we can see abundant loosely deposited quartz grain aggregates in 3D on unpolished surfaces (Figure 6B,C). These quartz particles include detrital, euhedral quartz microcrystals (Figure 6B), and siliceous spherical grain aggregates (i.e., quartz nanospheres) (Figure 6C). The filling or cementation by OM and clay is rarely seen between these particles, and a large number of interpores can be seen. These pores vary greatly in shape and size, with their sizes being generally less than one micrometer. On the whole, the observable pore size depends on the size of the quartz particles and the magnification of FIBSEM. It is important to note that the shape of a quartz interpore is commonly controlled by the arrangement of quartz particles. Rigid quartz grains could prop open pore spaces between them and thus prevent their collapse during compaction. Similar phenomena were also reported by Chalmers et al. [17]. In addition, abundant tiny quartz interpores (i.e., grain-edge pores) can be observed by TEM at a higher magnification (Figure 6D–F). These tiny grain-edge pores cannot be observed by FIBSEM. The TEM images suggest that quartz interpores are most likely formed between rigid quartz grains and are commonly concentrated around the rim of rigid grains, and these pores are mostly in the mesopore size fractions (5–20 nm in diameters). The pores in different particle contacting modes show different shapes ranging from linear to dendritic. 

#### 3.1.3. Aggregate-Hosted Pores

Apart from pore networks in the dispersed OM or minerals, an abundant complex pore space may also develop in the organic/inorganic aggregates. Several authors have suggested that aggregates can effectively preserve pore spaces [3,4,16]. For example, organic and clay grains are ductile and deform easily, and they can be strongly influenced by mechanical compaction during burial and later tectonism [3]. In contrast, quartz and pyrite grains are brittle and rigid, and they can largely resist compaction and structural deformation. Loucks et al. [16] recognized that rigid grains can act as the significant focal points where more ductile grains compact around them. Furthermore, Schieber [30] found that significant pore spaces can be preserved between rigid and ductile grains in the compaction-protected shadow adjacent to rigid grains. Visualizations from FIBSEM and TEM highlight the presence and morphological characteristics of the pore systems within these aggregates. A variety of pore networks exist in the shale samples, for instance, pores networks in clay-organic aggregates (Figure 7A,B), pyrite-organic aggregates (Figure 7C,D), clay-pyrite aggregates (Figure 7E), and clay-organic-pyrite aggregates (Figure 7F,G). Pore types in different aggregates possess different shapes and sizes. Such microstructural characteristics were produced by diagenetic processes and displayed a multiple-stage origin and diverse size/shape characteristic, which can be retraced to deposition, compaction, or cementation [16,26,31]. These four aggregate-hosted pores will be discussed in the next paragraphs. 

(I)Clay-Organic Aggregates

The organic particles were commonly found to be intimately associated with clay minerals, thus forming clay-organic aggregates. Figure 7A,B show FIBSEM and TEM images of the clay-organic aggregates from a Longmaxi sample, respectively. Large porosity cannot be observed within or between clay and organic layers in most microzones of the SEM image. Only a small part of the mesopores (pore size below 50 nm) are formed between the clay and organic layers. Note that abundant fine mesopores (pore size below 30 nm) indicated by red arrows in Figure 7A can be produced in organic layers. The pore size and morphology of these OM-hosted pores are restricted by the microstructure of the organic layers and adjacent clay layers. We believe that such clay-organic microstructures and their hosted nanopore system are the important evidence for the burial and compaction process of shale. In addition, these phenomena may indicate that these large organic pores are not like those observed in any of the previous examples, and they can occur along the clay grain boundary, suggesting that such pores can be selectively formed in OM because of the catalysis of clay plates. On the contrary, significantly smaller intrapores (pore size less than 5 nm, indicated by orange arrows) within clays, some mesopores (pore size less than 50 nm, indicated by green arrows) within organic particles, and interpores (pore size less than 5 nm, indicated by yellow arrows) between clay and organic particles can be found to be associated with clay-organic aggregates in the TEM image at a higher magnification (Figure 7B). From Figure 7A,B, we can infer that such pore networks occurring in related clay-organic aggregates were strongly influenced by mechanical compaction during deposition. Based on their size analysis, these pores may be the important adsorption spaces for methane. 

(II)Pyrite-Organic Aggregates

In addition to clay-organic aggregates, porosity was also observed in pyrite-organic aggregates. Figure 7C,D show the highly porous OM region formed between pyrite grains. Such aggregates contain pyrite and organic particles as small as 400 nm. In pyrite-rich areas within shales with high organic content, pyrite interpores appear to remain open and maintain connectivity during burial and compaction. Consequently, the dispersed migrated OM appears to fill most of such pores. The pressure shadows created by pyrite grains can significantly protect OM particles and the associated organic porosity from compactional collapse. The OM particles were encapsulated within open spaces between the pyrite grains. OM-hosted pores (less than 50 nm in size) are elliptical in shape and unevenly distributed within OM, and some organic porosities are not developed in some areas (Figure 7C). Similar pyrite-organic aggregates were also found in the TEM image (Figure 7D). In the TEM image, pyrite can be seen (black) with OM internal to groupings of individual grains of pyrite (Figure 7D). The TEM image analysis on the sample revealed that the pore networks are almost entirely restricted to the OM and pyrite aggregates. The significant porosity observed by both FIBSEM and TEM are composed of both (1) interpores (10~50 nm in size) between pyrites or between pyrite and OM, and (2) intrapores (~20 nm in size) within OM. Most of the pores are associated with the OM. 

(III)Clay-Pyrite Aggregates

A few small pyrite euhedral grains (~400 nm in width) occurred between some of the clay platelets (Figure 7E). Significant intrapores (~100 nm in size) developed within the clay particles and interpores (~250 nm in size) formed between clay platelets and pyrite grains can be observed, along with larger diameter irregular desiccation microcracks (Figure 7E). These pyrite euhedral grains commonly trapped within clay interlayers are the later diagenetic pyrites formed in shales. They may be produced by the following two processes: (1) the biotite particles were degraded and defoliated in pore waters with abundant free sulfide, and (2) the iron was released precipitated as pyrite along clay cleavage planes. This could suggest that the pyrite euhedral grains and pyrite framboids observed in the samples are not formed at the same time. Pyrite framboids belong to the products of early an early stage of diagenesis, while euhedral pyrites are formed in a later stage during burial. Intrapores occurring within clay particles could be strongly influenced by pyrite cementation. Such cements can significantly occlude the intrapore space. In addition, the interpores occurring between clay platelets and pyrite grains are generally present in smaller quantities and generally range between 200 and 300 nm in width (Figure 7E). 

(IV)Clay-Organic-Pyrite Aggregates

In the large clay domain of shale, it can also be seen that the later diagenetic pyrite particles and the migration asphaltene fill the clay-hosted pores together (Figure 7F,G). This phenomenon is very common in marine shales. The significant pore network observed by FIBSEM is composed of both (1) interpores (50~200 nm in size) between pyrite and clay or between pyrite and OM, and (2) intrapores (less than 50 nm in size) within OM and clay (Figure 7F,G). 

#### 3.1.4. Fracture-Related Pores 

As the significant nonmatrix pore and permeability network within shale gas systems, fracture-related pores can be classified as microchannels (Figure 8, indicated by green arrows) and microfractures (Figure 8, indicated by orange arrows) by their sizes and shapes. The existence, abundance, connectivity, origin, orientation, sealing properties, and proposed role in shale gas production of such pores have been commonly identified and documented by many studies [4,16,17,18,19,20]. It has been well established that fracture-related pores can play a key role in shale gas production. Moreover, Slatt and O’Brien [18] and Loucks and Reed [32] have documented them with reliable criteria. Slatt and O’Brien [18] noted that micro-channels and microfractures are linear nanometer- to micrometer-sized openings that could provide significant shale gas storage space and transport pathways in shale rocks. Loucks and Reed [32] defined natural microfractures in mudrocks as fractures with lengths less than 10 mm and widths less than 15 μm. Both studies highlighted the size, abundance, cement, and role of fracture-related pores in induced fracture propagation. The size range of the fracture-related pore recommended by Loucks and Reed [32] does not match the resolution of electron microscopy, thus the definition of fracture-related pore characterization that was proposed by Slatt and O’Brien [18] is used in this paper. 

Recently, the origin and importance of fracture-related pores in shale gas reservoirs have been the controversial subjects. They may have resulted from the process of internal (non-tectonic) factors and external forces (tectonic factors). Three important non-tectonic factors that can strongly influence the degree of fracture-related pore development in shale include: (1) fracturing the sample during the coring process, subsequent handling, and sample preparation (Figure 8A), (2) dehydration of clay minerals during burial and diagenesis (Figure 8B), and (3) thermal shrinkage (Figure 8C). The dehydration of clay minerals with abundant desiccation microfractures forming plays a significant role, followed by the thermal shrinkage. In the image in Figure 8B, we can observe that a microfracture (about 200 nm in width) extends between the large clay and rigid particles, which may be generated by the dehydration of clay minerals. The transformation of OM into hydrocarbons during thermal evolution creates abundant shrinkage microfractures due to overpressure. Artifacts can also be created during coring or post-coring, thus, how to identify them is a new challenge. Many researchers gradually suspected the importance of fracture-related pores, which is mainly due to the misinterpretation of artifact-related microfractures as natural microfractures [2,18]. Artifacts will strongly exaggerate the actual porosity and permeability and affect the shale gas reservoir evaluation. Therefore, it is significant to define what criteria should be used to separate natural microfractures from artifact-related microfractures. There is no doubt that the microfractures with non-sulfate-related cements are actually natural and not induced post-coring. Abundant natural microfractures can be closed by deposition of natural cements, and they can be significantly identified in thin sections by optical microscopes and electron microscopes. 

Natural microfractures are common, narrow, sealed with quartz, migration asphaltene, or calcite, and show complex cross-cutting relationships. Tectonic deformation continued after vein production, resulting in the formation of further veining [2]. Tectonic factors are the most important external causes of shale failure because of the accumulation and release of tectonic stresses during fault activities. Some significantly abundant microchannels and microfractures that are commonly found to be intimately associated with pyrite (Figure 8D) and calcite (Figure 8E), as well as quartz grains (Figure 8F), can be produced by tectonic deformation. In the TEM image of the sample, a microfracture with a width of about 100 nm can also be seen (Figure 8G). We infer that its genesis may be related to the water loss cracking of clay minerals. However, actually, no quantity of natural fracture-related pores that are not completely cemented has been found in the hundreds of shale samples, such as Barnett, Kimmeridge, Marcellus, Eagle Ford, Dunkirk, Longmaxi, and Qiongzhusi shales [2,4,16,18,30,32]. Some of our samples are taken from the shale-related fault zone, which provides the possibility to study the characteristics of natural microchannels and microfractures. Pore networks have been suggested to influence storage places and permeability pathways for gas migration. Among the different pore types, fracture-related pores appear to exert dominant controls upon the permeable pathways for fluid movement in shales. Firstly, both microchannels and microfractures are large enough to harbor relatively large quantities of hydrocarbon molecules, which are the significant storage space. Secondly, the intergrowth of microchannels and microfractures can be observed in some local microzones. The microfractures appear to develop around the grain boundaries, whereas the microchannels appear to develop within the grains. The microchannels can be seen to connect the adjacent microfractures, thus forming open fracture-related pore networks with good connectivity. Thirdly, both the microchannels and microfractures generate important flow networks to connect with each other and with the adjacent pores developed related to the OM and minerals, which can allow the escape of gas from small pores toward a large open microfractures. 

### 3.2. The Role of Pore Architecture in Gas Storage and Migration

Several authors have noted that the pore architecture has a significant control over gas storage and migration due to its various types, sizes, morphologies, arrangements, and locations, as well as its connectivities. For example, Ambrose et al. [33] demonstrated that the quantity of sorbed gas that is stored on the pore walls becomes significant when the pore sizes are less than 50 nm. If the pore sizes are less than 10 nm, the gas absorption becomes a very strong effect. These small pores are considered to be the dominant sorption sites. It is important to note that only pores related to OM, clays, and clay-organic aggregates can provide abundant adsorption sites, whereas other mineral-related pores can provide few adsorption sites. If the pore sizes are more than 50 nm, the quantity of free gas that is stored within the pores becomes significant. The pore size distribution (PSD) and associated gas storage characteristics and connectivity of different pore types are summarized in Figure 9. Figure 9 illustrates how they distribute the pore size according to their classifications. We can clearly see that the microfractures have the largest pore size and the best pore connectivity, while the carbonate solvopores have the lowest pore connectivity. Note that OM-hosted pores and clay intrapores have a wider PSD than other pore types, indicating a complex gas occurrence mechanism within them on the micro- and nanoscales. It is generally known that the hydrocarbon molecule is only 0.38 nm in size and an average OM-hosted pore or clay intrapore of tens of nanometers can fit approximately 100 molecules, that is to say, these pores are all large enough to store hydrocarbon molecules (e.g., sorbed gas or free gas). As shown in Figure 7A,B, a significant part of the OM-hosted pores, clay intrapores, and aggregate-hosted pores developed within the samples are sufficiently small. The walls of such pores are perforated with smaller sized pores. As a result, hydrocarbon molecules stored in these pores should be dominated by sorbed gas. In addition, as illustrated in Figure 4, Figure 5 and Figure 6, a significant part of the mineral-hosted pores, such as carbonate solvopores, pyrite interpores, and quartz interpores are sufficiently large and can contain dominantly free gas. 

Published observational studies suggest that if different types of pores are present in sufficient quantities and can connect together, they will create an effective pore network [16]. Moreover, several studies have shown that the larger the proportion of OM-hosted pores, mineral-hosted interpores, and fracture-related pores, the higher the effectiveness of the pore system is [15,16,17,18]. In contrast, the larger the proportion of mineral-hosted intrapores, the lower the effectiveness of the pore network is, which is a result of their particular architectures [16], such as closure and smaller pore throats. These conclusions are well known. As shown in our FIBSEM and TEM images, major differences in the pore geometry can be seen. The predominant pore types are OM-hosted pores and clay intrapores. These pore types are open and abundant and can provide potential gas storage spaces and gas transport pathways. Fracture-related pores are also common in the samples and can also provide significant migration pathways for shale gas. Carbonate solvopores, pyrite interpores, and quartz interpores, as well as aggregate-hosted pores, are relatively rare. The open pores with considerable amounts of components, such as microfractures, are generally better connected to one another than the closed pores, and they can contribute more to the effective pore network than the other pore types can. However, some closed pores are nearly totally cut off from the whole effective pore network. Therefore, the preponderance of open pore types can offer a high-quality matrix pore system. 

In this study, pore types were classified into four classes, each with specific PSD, pore abundance, pore connectivity, porosity, permeability, and reservoir quality, as well as their contribution to the whole effective pore network (Figure 10). Unlike the high porosity observed within the mineral-hosted pores and fracture-related pores or the medium porosity observed within OM-hosted pores, the porosity developed in the aggregate-hosted pores is very low in abundance. OM-hosted pores and fracture-related pores contribute more to the effective pore systems than other pore types do. The excellent reservoir quality is produced by OM-hosted pores and fracture-related pores, whereas poor reservoir quality may come from aggregate-hosted pores. 

### 3.3. The Role of Thermal Maturity and OM Composition in Variation of Organic Porosity 

OM-hosted pores are associated with the generation of hydrocarbons in situ from OM [15,16,21,22]. The importance of pores within OM in the microstructures of shale gas reservoirs has attracted unprecedented attention by many oil and gas geologists over the past 15 years [15,18]. The discovery of OM-hosted pores by using SEM-based or TEM-based imaging techniques could spark two key questions concerning their evolution and nature: (1) are the OM-hosted pores created by the transformation of kerogen into bitumen, oil, and gas during thermal maturation of OM, and (2) what is the function of OM composition in the development of OM-hosted pores? The answers to the above two questions are key to discussing the preliminary model of organic porosity evolution that is controlled both by thermal maturation and the OM types and understanding the heterogeneity in the character of OM composition in shale rocks. 

#### 3.3.1. Control of OM-Hosted Pores Evolution by Thermal Maturity

The debate about the formation and development of OM-hosted pores in organic-rich shales of varying maturity is significant because OM-hosted pores are widely recognized as the principal gas storage space and migration pathways in gas shales. Recent studies of the Mississippian Barnett shale of the Fort Worth Basin, Texas, show a strong correlation between OM nanopores and vitrinite reflectance, suggesting that thermal maturation is a key controlling factor promoting the generation of OM-hosted pores [15], which can be strongly supported by the absence of organic nanoporosity from their lower-thermal-maturity shale samples and the abundance of organic nanoporosity in the more thermally mature samples. This relationship is consistent with observations made by Curtis et al. [34] on eight Woodford shale samples (Late Devonian-Early Mississippian, their mean random vitrinite reflectance values ranging from 0.51% to 6.36% *R*_o_) using the SEM. They suggested the development of organic porosity with increasing kerogen maturity. Furthermore, Bernard et al. [35] used a combination of synchrotron-based scanning transmission X-ray microscopy (STXM) and X-ray absorption near-edge structure (XANES) spectroscopy to indicate that OM-hosted pores can only be produced within OM in the gas window. However, other scholars thought that such arguments no longer hold up. For example, Reed and Ruppel [36] found that the oil-window maturity shale samples can also contain abundant OM-hosted nanopores, which is contrary to the conclusion of Bernard et al. [35]. Collectively, these published data suggest that the formation of OM-hosted pores is the result of thermal maturation and conversion of kerogen. However, at which stage the organic porosity first occurs, it is still not well understood. Additional investigations are needed to fully understand the control of thermal maturity on OM-hosted pores development. 

Based on the results of the published literature and this present study, we conclude a possible development model of organic porosity within shale samples of varying maturity. To help understand the nature of organic particles during different stages of thermal evolution and the associated organic pore networks, an idealized history of OM-hosted pore development with increasing thermal maturity and later tectonism is discussed below and illustrated in Figure 11. With increasing thermal maturity, thermal stress causes the kerogen to crack and to transform first into (1) bitumen and then into (2) petroleum fluids, and finally (3) gas. During the first transformation stage (thermal maturity generally below 0.6% *R*_o_), kerogen to bitumen, a small number of available organic nanopores can be firstly produced with small sizes, a small quantity, and poor connectivity. Such nanopores are isolated. During the second transformation stage (*R*_o_~0.6–1.4%, in the oil window), bitumen to petroleum fluids, considerable amounts of oil fluids can be generated and migrated into adjacent large mineral-hosted interpores and microfractures in the shale. Consequently, these migrated fluids may significantly reduce the volumes of the mineral-hosted interpores and microfractures. In addition, organic nanopores are weakly developed. Such nanopores show bubble-like outlines. During the last transformation stage (thermal maturities generally greater than 1.4% *R*_o_, in the gas window, such as the wet gas window with ~1.4% *R*_o_ and the dry gas window with ~2.1% *R*_o_), petroleum fluids to gas, a large number of organic irregular spongy pores are significantly generated with large sizes and good connectivity, which can create an effective pore network within OM. After these three transformation stages, organic porosities may be strongly compressed due to the effect of later tectonic stress, compaction, or the release of local pressure, thus they collapse, resulting in an extreme reduction of OM-hosted pore volume. This finding is in agreement with a recent study [37]. The researchers suggested that considerable amounts of OM-hosted pores cannot be well preserved after the processes of oil or gas generation and further concluded that mineral-hosted interpores and intrapores play a key role in the storage of hydrocarbons. However, they did not point out the specific reasons for the reduction in organic porosity. 

We speculate that tectonism may be a significant factor that influences the destruction of organic pore spaces. However, in this process, a large number of rigid mineral particle skeletons can play the role of buffering and resisting tectonic stress (i.e., mineral sheltering). A large number of pressure shadows are formed locally in the shale matrix, for example, the microstructure of the OM-hosted pore in the quartz interpores is usually well preserved. Unlike the shale gas reservoirs of North America, most organic-rich shale layers in China have experienced multi-stage tectonic events after the end of hydrocarbon generation. Shale reservoirs have been generally destroyed by tectonic deformation and natural fractures. On the one hand, some ductile particles such as OM and clays are mechanically unstable and deform and shrink easily, making the pore volume reduce. On the other hand, a large number of natural microfractures and mineral-hosted interpores are produced, which leads to the release of hydrocarbons in the reservoirs. Some early work on OM-hosted pores suggested that preservation of these pores is dependent of the pore pressure and stress. It is conceivable that natural microfractures can strongly reduce the pore pressure in the OM. Therefore, our study supports the idea that a decrease in organic porosity volume at high maturity may be related to tectonic compression and compaction. 

#### 3.3.2. Control of OM-Hosted Pores Evolution by OM Composition 

Numerous workers have recently suggested that not all organic particles appear to be prone to the development of OM-hosted pores at a high thermal maturity. For example, Loucks et al. [16] suggested that type II kerogen appears to be more prone to the formation of OM-hosted pores than other types. In addition, they also suggested that limited data and samples, as well as residual oil that occurred within the organic porosity may strongly cause the mistaken identification of OM-hosted pores. Furthermore, Mastalerz et al. [38] also support this idea. The authors demonstrated that organic pore spaces can be strongly reduced at some levels during maturation. They interpreted such phenomenon as the result of pore filling by incompressible fluid, such as water, oil, or solid bitumen, which can significantly block the pore throats, thus reducing the available open pore space. In addition, Milliken et al. [31] cautiously pointed out that the development of OM-hosted pores within Marcellus shales does not display a dependence on thermal maturity in the gas window, but rather on the heterogeneity of OM types. Formation of OM-hosted pores may also be dependent on OM compositions during thermal evolution, because the hydrocarbon generation temperatures, the kerogen decomposition rates, and the chemical compositions vary for the organic particles in different source rocks [39,40]. Therefore, the importance of OM types in the generation of OM-hosted pores has also been a controversial subject. We agree with some recent investigations in claiming that the organic porosity can be altered during maturation by OM composition [16,34]. We do acknowledge that no development of organic porosity in some organic particles at high maturity may be related to OM types. This view is further illustrated by Figure 12 which presents two distinguished regions in a large organic particle from a Longmaxi shale sample with a thermal maturity of about 1.9% *R*_o_. A significant difference in the development of the OM-hosted pores can be observed in these two regions. In region A in Figure 12, numerous organic slit-like pores with sizes ranging from a few nanometers to hundreds of nanometers can be observed. These pores are indicated by white arrows on this FIBSEM image. This is in contrast to the mostly smaller and poorly developed OM-hosted pores which can be seen in region B. Similar microstructures were also reported by Curtis et al. [34] in Figure 6 of their study. Their study supports the ideas that (1) since both regions have experienced the same thermal history, there must be something fundamentally different about the OM types with different macerals and the OM composition could complicate the prediction of the development of organic porosity with increasing thermal maturity, and (2) the evolution of secondary OM-hosted pores can be significantly affected by the maceral types (i.e., fusinite, plant cell lumens, alginite, liptinite, vitrinite, inertinite, or kerogen and bitumen). 

This study visually studies the role of thermal maturity and OM composition in the variation of organic porosity. FIBSEM and TEM imaging provide direct information about the nature of OM-hosted pores. We acknowledge that our shale samples are limited. However, additional shale samples from different shale formations, ages, and areas in the world, which can reflect various vitrinite reflectance values and OM types could collectively help to better understand the impact of thermal maturity and OM composition on organic porosity development. 

## 4. Conclusions

This investigation presents microstructural characterization of the pore type, pore morphology, pore size, and pore network of prospective organic-rich marine shales in the Sichuan Basin, South China, using a combined application of FIBSEM and TEM. From this contribution, we conclude that:(1)FIBSEM and TEM imaging tools can be used as complementary methods to other prominent techniques such as HP, LPGA, MIP, and NMR to provide intuitive observations for visualizing microstructures and to improve our understanding of the micro-nanoscale petrophysical characteristics of organic-rich shales.(2)Multiple matrix-related pore types were identified, including OM-hosted pores, mineral-hosted pores, aggregate-hosted pores, and fracture-related pores.(3)OM-hosted pores are universal and range in size from below 1 nm to hundreds of nanometers. They are not homogeneously developed and distributed, which is mainly determined by thermal maturity and OM composition.(4)Mineral-hosted pores are defined by mineral frameworks and subdivided into clay intrapores, carbonate solvopores, pyrite interpores, and quartz interpores, and range in size from less than 1 nm to more than several microns.(5)Aggregate-hosted pores are predominantly associated with clay-organic aggregates, pyrite-organic aggregates, clay-pyrite aggregates, and clay-organic-pyrite aggregates, although other minerals (e.g., quartz, feldspar, and calcite) may show aggregate effects as well. The most common aggregate-hosted pore networks are defined by clay-organic aggregates, and the pores are largely developed between the clay and organic layers and may be the important adsorption spaces for methane.(6)Fracture-related pores include microchannels and microfractures of various sizes and shapes, and they can play a key role in providing hydrocarbon migration pathways.(7)FIBSEM and TEM show direct evidence that OM-hosted pores and fracture-related pores contribute more to the effective pore network and the excellent reservoir quality, whereas poor reservoir quality may come from aggregate-hosted pores.

## Figures and Tables

**Figure 1 nanomaterials-12-04135-f001:**
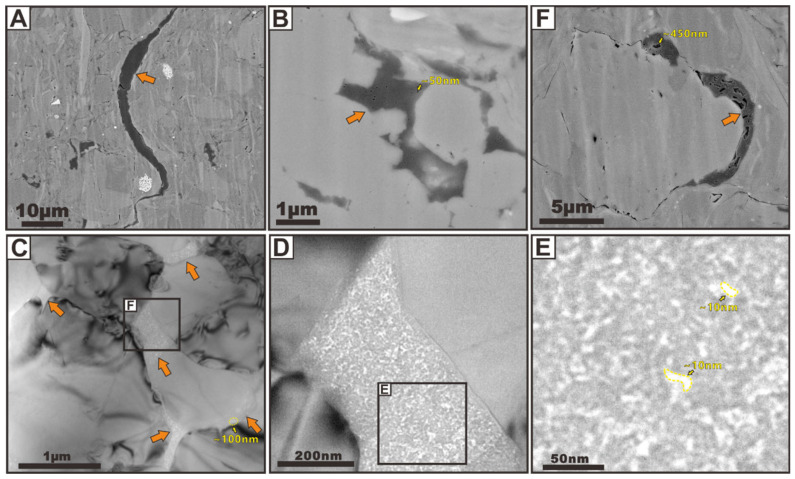
(**A**) SEM images of FIB-milled surface showing secondary migrated OM in microfractures. (**B**) FIBSEM image showing secondary migrated OM in large interpores. (**C**–**E**) TEM images showing secondary migrated OM in large interpores and their related OM-hosted pores. The images (**C**–**E**) are modified after Zhu et al. [20]. (**F**) FIBSEM image showing secondary migrated OM in large grain-rim dissolution pores. The secondary migrated OM are indicated with orange arrows, and the OM-hosted pores are indicated with yellow arrows.

**Figure 2 nanomaterials-12-04135-f002:**
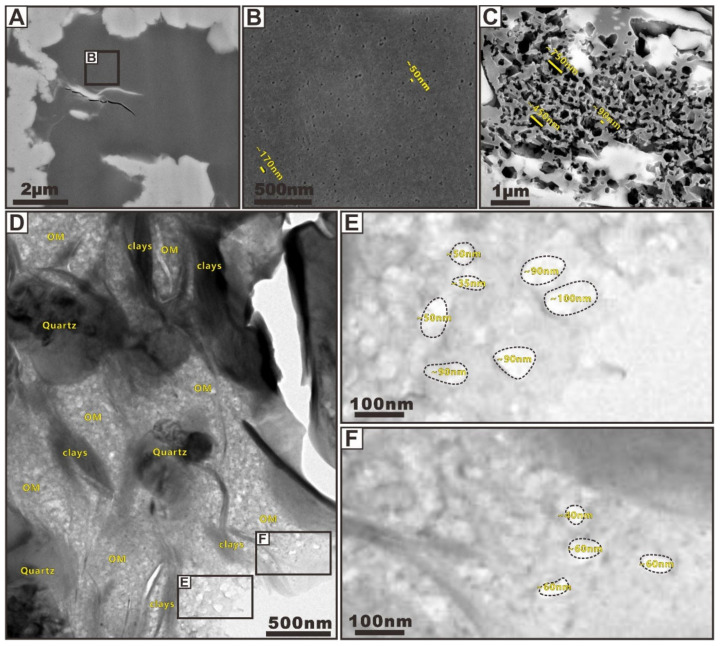
(**A**–**C**): SEM images of FIB-milled surface showing depositional in-place OM and their related OM-hosted pores. (**D**–**F**): TEM images showing depositional in-place OM and their related OM-hosted pores. The images (**D**–**F**) are modified after Zhu et al. [3].

**Figure 3 nanomaterials-12-04135-f003:**
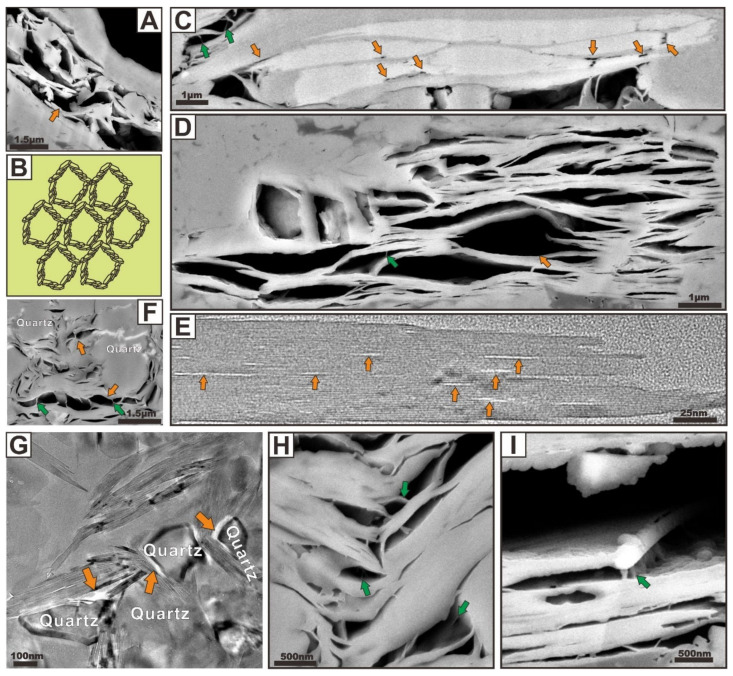
(**A**) FIBSEM image showing typical “cardhouse” of clay flakes. (**B**) The model showing the “cardhouse” of clay flakes. (**C**,**D**) FIBSEM images showing slit clay intrapores that are typically reduced in abundance and size by compaction. (**E**) TEM image showing slit clay intrapores that are typically reduced in abundance and size by compaction. (**F**) FIBSEM image showing typical jagged clay intrapores in the strain shadows of clasts. The image (**F**) is modified after Zhu et al. [19]. (**G**) TEM image showing typical jagged clay intrapores in the strain shadows of clasts. The image (**G**) is modified after Zhu et al. [3]. (**H**,**I**) FIBSEM images showing lath-like illitic nanostructures, which could provide a large number of methane adsorption sites. The image (**H**) is modified after Zhu et al. [19]. The image (**I**) is modified after Zhu et al. [14]. Typical clay intrapores are indicated with orange arrows, and lath-like illitic nanostructures are indicated with green arrows.

**Figure 4 nanomaterials-12-04135-f004:**
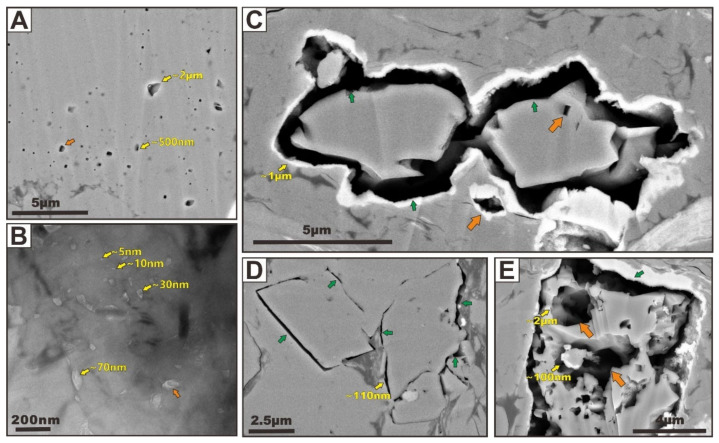
(**A**) FIBSEM image of typical carbonate solvopores because of partial dissolution showing bad microstructural connectivity. (**B**) TEM image of typical carbonate solvopores because of partial dissolution showing bad microstructural connectivity. (**C**) FIBSEM image showing a large channel and vuggy carbonate solvopores showing good microstructural connectivity. The image (**C**) is modified after Zhu et al. [19]. (**D**) FIBSEM image showing typical grain-rim carbonate solvopores. (**E**) FIBSEM image showing the subhorizontal large solvopores through a carbonate grain exposes abundant nanopores developed at the inner surface of pore walls. The image (**E**) is modified after Zhu et al. [19]. Typical carbonate solvopores produced by partial dissolution are indicated with orange arrows, and grain-rim carbonate solvopores are indicated with green arrows.

**Figure 5 nanomaterials-12-04135-f005:**
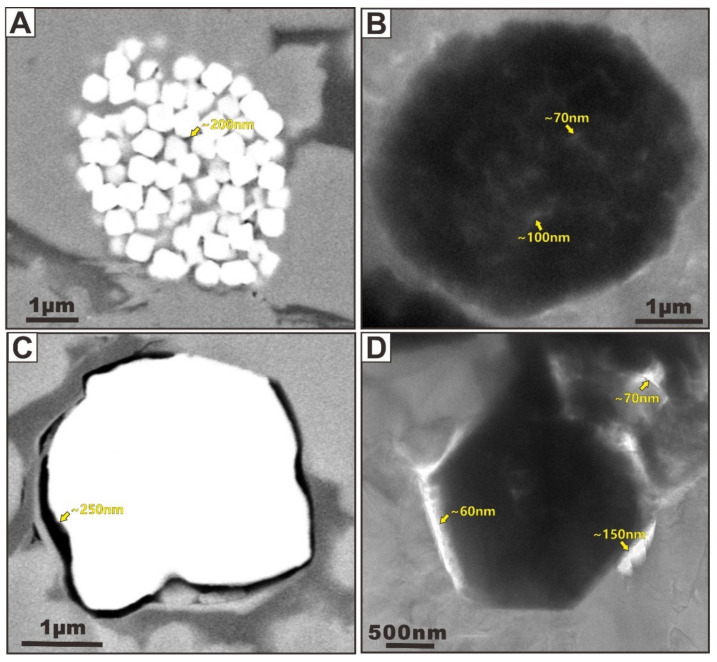
FIBSEM (**A**) and TEM (**B**) images of typical pyrite framboids showing abundant pyrite interpores between pyrite grains. FIBSEM (**C**) and TEM (**D**) images of typical pyrite crystals showing abundant pyrite interpores between pyrite grain and other rigid grains. The pyrite interpores are indicated with yellow arrows.

**Figure 6 nanomaterials-12-04135-f006:**
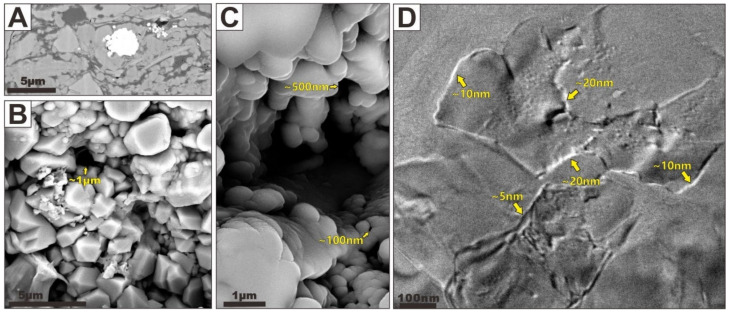
FIBSEM (**A**–**C**) and TEM (**D**) images showing typical quartz grains and quartz interpores in shale. The images (**A**–**C**) are modified after Zhu et al. [20]. The quartz interpores are indicated with yellow arrows.

**Figure 7 nanomaterials-12-04135-f007:**
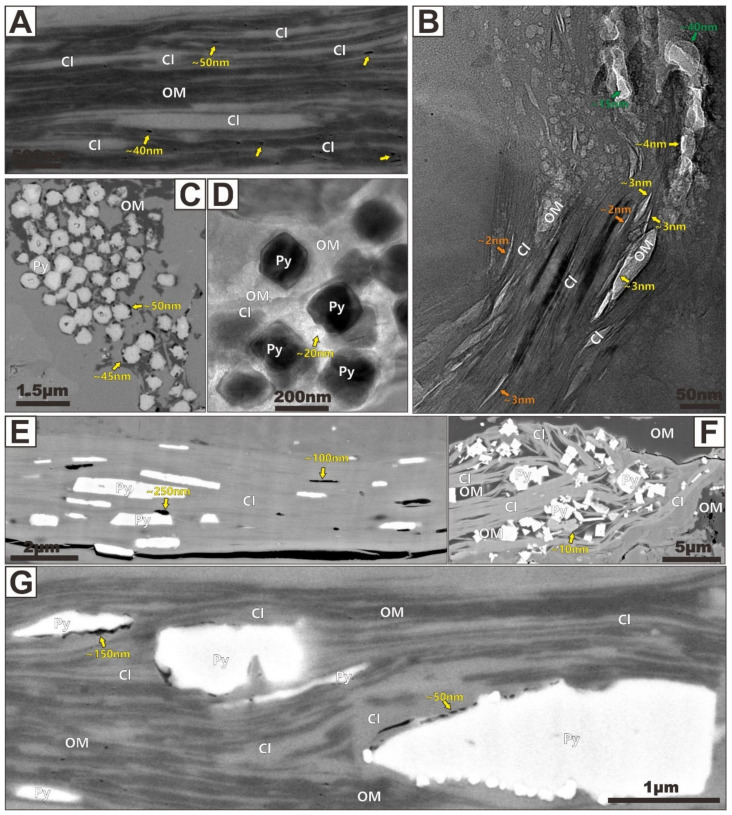
FIBSEM (**A**) and TEM (**B**) images showing typical clay-organic aggregates and their hosted pore system. The image (**A**) is modified after Zhu et al. [3]. The image (**B**) is modified after Zhu et al. [14]. The smaller intrapores (pore size less than 5 nm) within clays are indicated by orange arrows, the mesopores (pore size less than 50 nm) within organic particles are indicated by green arrows, and the interpores (pore size less than 5 nm) between clay and organic particles are indicated by yellow arrows in the image (**B**). FIBSEM (**C**) and TEM (**D**) images showing typical pyrite-organic aggregates and their hosted pore system. (**E**) FIBSEM image showing the typical clay-pyrite aggregates and their hosted pore system. (**F**,**G**) FIBSEM images showing the typical clay-organic-pyrite aggregates and their hosted pore system. The image (**F**) is modified after Zhu et al. [19]. The image (**G**) is modified after Zhu et al. [3].

**Figure 8 nanomaterials-12-04135-f008:**
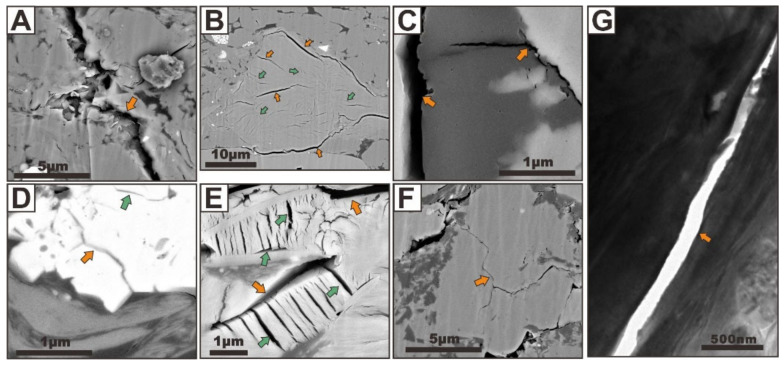
(**A**) FIBSEM image showing microfracture produced by sample preparation. (**B**) FIBSEM image showing microfracture because of dehydration of clay minerals. (**C**) FIBSEM image showing microfracture due to thermal shrinkage of organic particles. FIBSEM images showing microchannels and microfractures that are commonly found to be intimately associated with pyrite (**D**), calcite (**E**), and quartz grains (**F**). (**G**) TEM image showing a microfracture with a width of about 100 nm because of the dehydration of clay minerals. The microfractures are indicated with orange arrows, and the microchannels are indicated with green arrows.

**Figure 9 nanomaterials-12-04135-f009:**
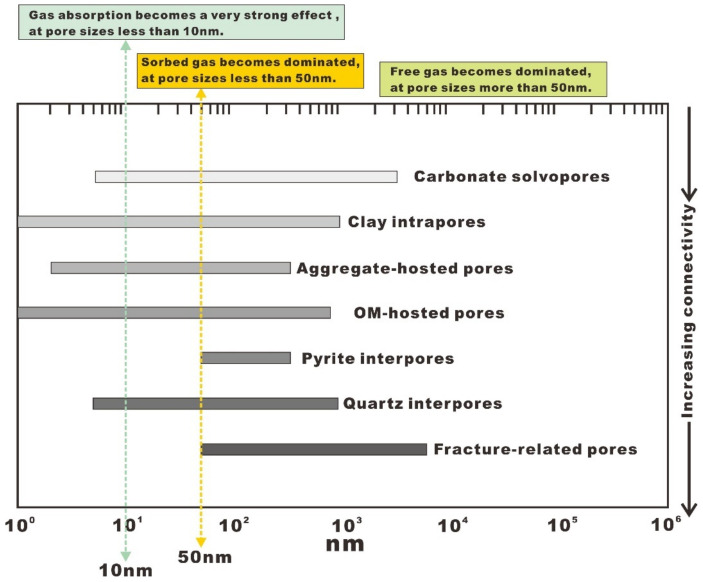
Aperture distribution range of each pore class based on visual observations.

**Figure 10 nanomaterials-12-04135-f010:**
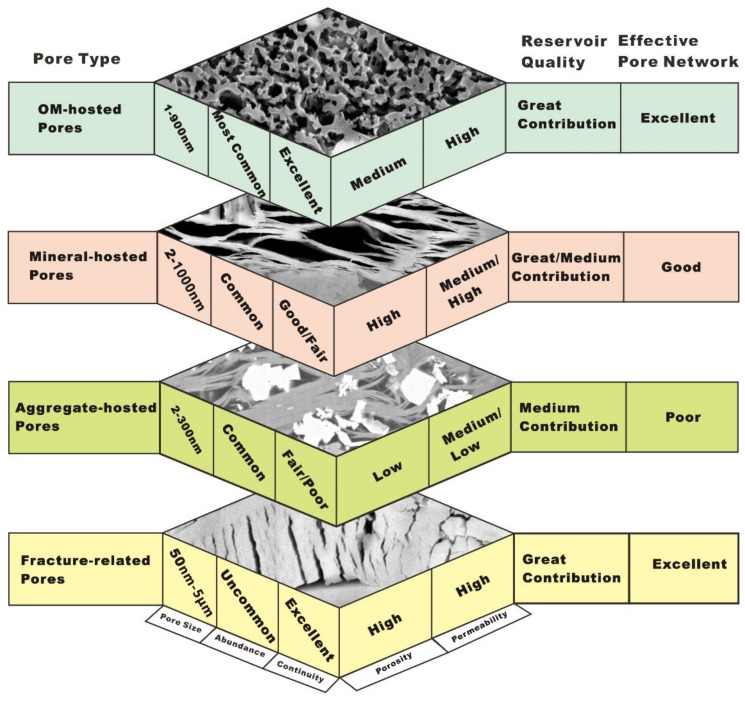
The main attributes of the pore types including pore size, pore abundance, pore connectivity, porosity, permeability, reservoir quality, and their contribution to the whole effective pore network within each pore class.

**Figure 11 nanomaterials-12-04135-f011:**
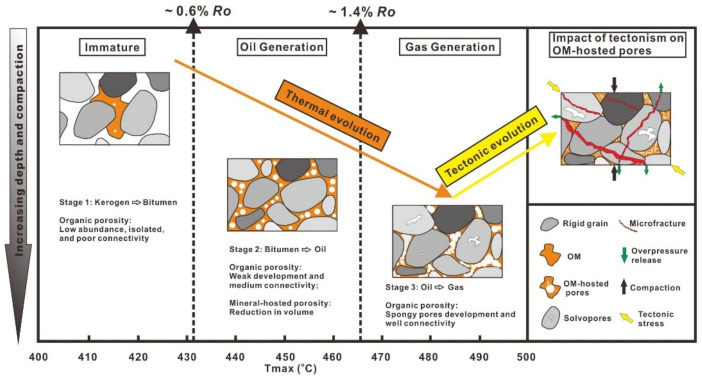
Schematic diagram showing the possible pattern diagram of development of OM-hosted pore network with increasing thermal maturity and later tectonism.

**Figure 12 nanomaterials-12-04135-f012:**
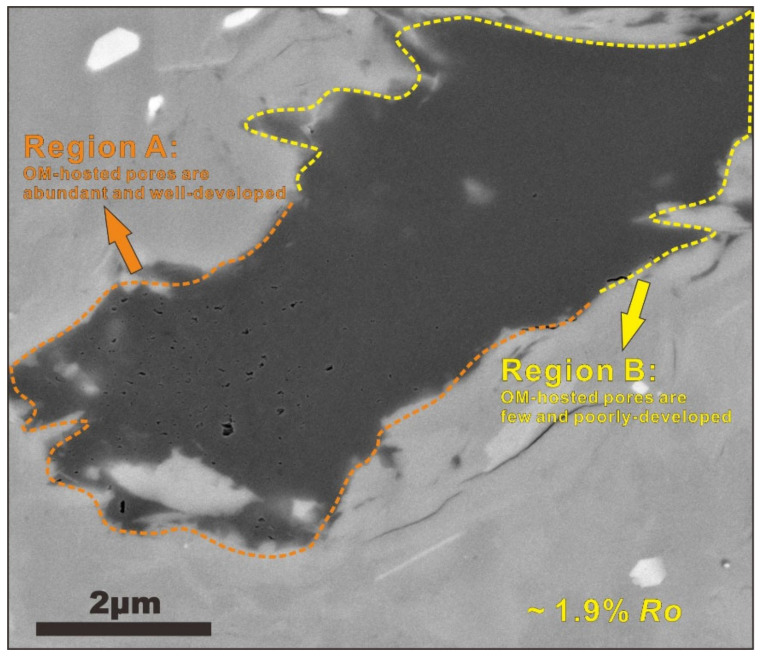
FISSEM image of Longmaxi shale sample with a thermal maturity of OM of 1.9% *R*_o_. The organic region on the bottom left shows numerous OM-hosted pores whereas the organic particles on the top right show no pores.

## Data Availability

Not applicable.

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
