# Peer review of "Microstructural Analysis of Organic-Rich Shales: Insights from an Electron Microscopic Study by Application of FIBSEM and TEM"

_nanomaterials, 2022, doi:10.3390/nano12234135_

Round 1

Reviewer 1 Report

The manuscript "Microstructural analysis of organic-rich shales: Insights from an electron microscopic study by application of FIBSEM and TEM" from J. Han et al. has interesting results concerning the analysis by FIBSEM and TEM of the pore networks in shales. In my opinion the manuscript is suitable for publication after some modifications:

- The abstract it is too long it has many information that should be in the introduction. It should describe what was done in the work and the problem they want to address. 

- What is the limitation of the methods the authors used; it should be described.

- Other techniques such as CT scan is much used in the study of geological samples although the resolution is not so high, but it may give interesting data also. 

Author Response

Respond to reviewer #1

We really appreciate your contribution to our manuscript. Your suggestions make the revised manuscript easier to be read and highlight our main conclusions.

Issue 1: The abstract it is too long it has many information that should be in the introduction. It should describe what was done in the work and the problem they want to address.

Response:

Thanks for your advice. Sorry for our mistake. We have corrected it. We simplified the text of the abstract, highlighting the innovation and main achievements of the manuscript. The specific modifications are marked in our modified version with the modification mode.

Issue 2: What is the limitation of the methods the authors used; it should be described.

Response: Good suggestion. We have indeed overlooked this detail. As we all know, any research methods to characterize shale pore structure has some drawbacks or limitations, such as low-temperature gas adsorption experiment and high-pressure mercury intrusion experiment. These two experiments can only test and calculate the structure of open pores, and can only obtain quantitative data, which cannot be visualized. In contrast, with the continuous development of SEM and TEM, the micro- to nanopore structure of shale can be directly displayed to scientists. However, these electron microscopy methods also have certain limitations, such as small field of view, poor representativeness, and cannot solve the problems of shale microstructural heterogeneity. Therefore, the best solution at present is to jointly apply gas adsorption experiment, mercury intrusion experiment and electron microscope technology, so as to characterize the shale microstructure more comprehensively and accurately. The specific modifications are marked in our modified version with the modification mode in the chapter 2.3.

Issue 3: Other techniques such as CT scan is much used in the study of geological samples although the resolution is not so high, but it may give interesting data also.

Response: Thank you very much for your advice. As you mentioned, CT technology, like SEM and TEM, is a significant shale microstructural characterization technology developed rapidly in recent years. It can not only characterize the two-dimensional structure, but also provide very accurate three-dimensional spatial structure information. However, the current resolution of this technology is poor. Pore and microfracture structures with dimensions of hundreds of nanometers can hardly be obtained by CT technology. Because of its resolution, we did not consider it at the beginning of organizing the manuscript. Anyway, thank you for your suggestions. We will adjust our later work and apply the CT technology you mentioned to our research objects.

Reviewer 2 Report

My comments listed in the attached file, This is a well-written paper.

Author Response

Respond to reviewer #2

We really appreciate your contribution to our manuscript. Your suggestions make the revised manuscript easier to be read and highlight our main conclusions.

Issue 1: Several aspects of the figures need improvement. One problem is that some scale bars lack readability (examples: Figure 1A, D, H). A more general issue is the use of yellow and white font colors on light-colored images, where the characters have low visibility. Many figures contain arrow markings that are not explained. Figure 3 is an example where the arrows are explained. Most other figures lack explanations for the arrows. Many figures include nm values with no explanation of their meaning. Are they meant to indicate pore diameters? The captions need to be improved.

Response: Thanks for your advice. Sorry for our mistake. We have corrected them. We have revised and improved the scale of the pictures and the information indicated by the arrows. The scale is uniformly changed to black. The arrow information is explained in the title and the original text. Very good suggestions for details, thanks! The specific modifications are marked in our modified version.

Issue 2: Figures 9-12 are not in sharp focus, I presume this may simply be the result of the use of low resolution files used for PDF version created for review purposes.

Response: Yes, we provide editable high-resolution bitmaps in figure word. The resolution can meet the standards of journal publishing. The picture is distorted due to the conversion of word to pdf file.

Issue 3: My suggestions for grammar modifications all involve very minor issues. Line 40: delete “the” from “the Lower Cambrian and Lower Silurian shales”.

Response: Thank you for your suggestion. However, we suggest that the Lower Cambrian and Lower Silurian belong to proper nouns. Should “the” be added before them?

Issue 4: Line 43: “ever increasing” should be “ever-increasing”;

Response: Good suggestion. We have modified it.

Issue 5: Line 57: “Assessments of pore structures (morphology, size, connectivity, etc.) are typically made by indirectly attempt analysis such as helium porosity…” The wording needs to be changed. Do you mean “attempting indirect analysis”?

Response: Thank you for your suggestion. We will modify it as follows: Some attempt analyses such as helium porosity (HP), low-pressure gas adsorption (LPGA), mercury injection pressure (MIP), high-pressure CH4 isotherm analysis, and nuclear magnetic resonance (NMR) spectroscopy are used to assess pore structures (morphology, size, connectivity, etc.).

Issue 6: Line 61: “Thus, the electron microscopy (EM) technique has long been…” could be simplified to “Electron microscopy (EM) has long been….

Response: Good suggestion. We have modified it.

Issue 7: Line 91: “A large number of previous studies…” is followed by only four citations. Maybe better wording would simply be “Previous studies….”

Response: Good suggestion. We have modified it.

Issue 8: Line 518: “grain” should be “grains”

Response: Very good suggestions for details. I'm sorry we made a grammatical mistake. We have corrected it.

Issue 9: Line 521: “defoliated” doesn’t seem like the right word. Maybe “delaminated”?

Response: Thank you for your suggestion. This term ‘defoliated’ should be applied to chemical changes of flaky particles or minerals, such as biotite and muscovite. Therefore we suggest this word can be retained.

Issue 10: Line 527: “during burial” and “later in burial” lack descriptive clarity. “During burial” presumably is meant to describe an early stage of diagenesis that produced framboids, followed by a later episode that produced euhedral pyrite crystals.

Response: Good suggestion! We will modify it as follows: Pyrite framboids belong to the products of early an early stage of diagenesis, while euhedral pyrites are formed in a later stage during burial.